# Total syntheses of (−)-macrocalyxoformins A and B and (−)-ludongnin C

Zichen Cao[1,2,4], Wenxuan Sun[2,3,4], Jingfu Zhang[2], Junming Zhuo[2], Shaoqiang Yang[2], Xiaocui Song[2], Yan Ma[2,3], Panrui Lu[2,3], Ting Han [2,3] & Chao Li [2,3] ✉

The complex and diverse molecular architectures along with broad biological activities of *ent*-kauranoids natural products make them an excellent testing ground for the invention of synthetic methods and strategies. Recent efforts notwithstanding, synthetic access to the highly oxidized enmein-type *ent*-kauranoids still presents considerable challenges to synthetic chemists. Here, we report the enantioselective total syntheses of C-19 oxygenated enmein-type *ent*-kauranoids, including (−)-macrocalyxoformins A and B and (−)-ludongnin C, along with discussion and study of synthetic strategies. The enabling feature in our synthesis is a devised Ni-catalyzed decarboxylative cyclization/radical-polar crossover/*C*-acylation cascade that forges a THF ring concomitantly with the *β*-keto ester group. Mechanistic studies reveal that the *C*-acylation process in this cascade reaction is achieved through a carboxylation followed by an in situ esterification. Biological evaluation of these synthetic natural products reveals the indispensable role of the ketone on the D ring in their anti-tumor efficacy.

Various enzyme-mediated C−H oxidations, C−C bond cleavages, and fragmentations of tetracyclic *ent*-kaurene (**1**, Fig. 1a) generate a truly incredible array of *ent*-kaurane diterpenoids[1–3]. To date, more than 1000 members of this class have been isolated from diverse *Isodon* species[3]. The structural diversity also imbues this class of diterpenoids with a broad spectrum of bioactivities (e.g., antitumor, antibacterial properties)[1–3]. Accordingly, these diterpenoids have been conceptualized as an excellent testing ground for demonstrating new synthetic strategies and methods, and a number of remarkable total syntheses have been achieved over the past decades[4–12].

Enmein-type *ent*-kauranoids are among the most highly oxidized members of the *ent*-kauranoid family[1]. To date, more than 90 enmein-type diterpenoids have been identified. Biosynthetically, they are thought to be generated from **1** through a C6−C7 oxidative cleavage, a C1−C7 lactonization, and various C−H oxidations (Fig. 1a). Multiple properties, including the high oxidation state, the twist-boat conformation of the B lactone ring, and the bridged boat-conformation of

the C ring represent huge challenges to chemical synthesis efforts. Indeed, only 3 enmein-type *ent*-kauranoids (Fig. 1b) have been synthesized over the past half-century: the Fujita group completed the relay synthesis of enmein (**2**) in 1974 over 42 steps[13,14]. In 2018, the Dong group reported a divergent total synthesis of three enmein-type diterpenoids: (−)-enmein (**2**), (−)-isodocarpin (**3**), and (−)-sculponin R (**4**)[15]. Despite these elegant studies, total synthesis of C-19 oxygenated enmein-type *ent*-kauranoids such as (−)-macrocalyxoformins A (**5**) and B (**6**)−which possess one more synthetic challenging C4 quaternary stereocenter as compared to **2**–**4**−have not been achieved[16–19].

As part of our ongoing research program aimed at the collective total synthesis of bioactive and structurally diverse *ent*-kauranoids, leveraging meticulously designed radical cascade reactions[20,21], we pursued a radical cascade approach to synthesize the distinct fused A/E1/E2 ring system found in the unexplored highly oxidized C19-oxygenated enmein-type *ent*-kauranoids, such as (−)-macrocalyxoformins A (**5**) and B (**6**), as well as (−)-ludongnin C (**7**). Despite the

[1]School of Life Sciences, Peking University, 100871 Beijing, China. [2]National Institute of Biological Sciences, 102206 Beijing, China. [3]Tsinghua Institute of Multidisciplinary Biomedical Research, Tsinghua University, 100084 Beijing, China. [4]These authors contributed equally: Zichen Cao, Wenxuan Sun. ✉e-mail: lichao@nibs.ac.cn

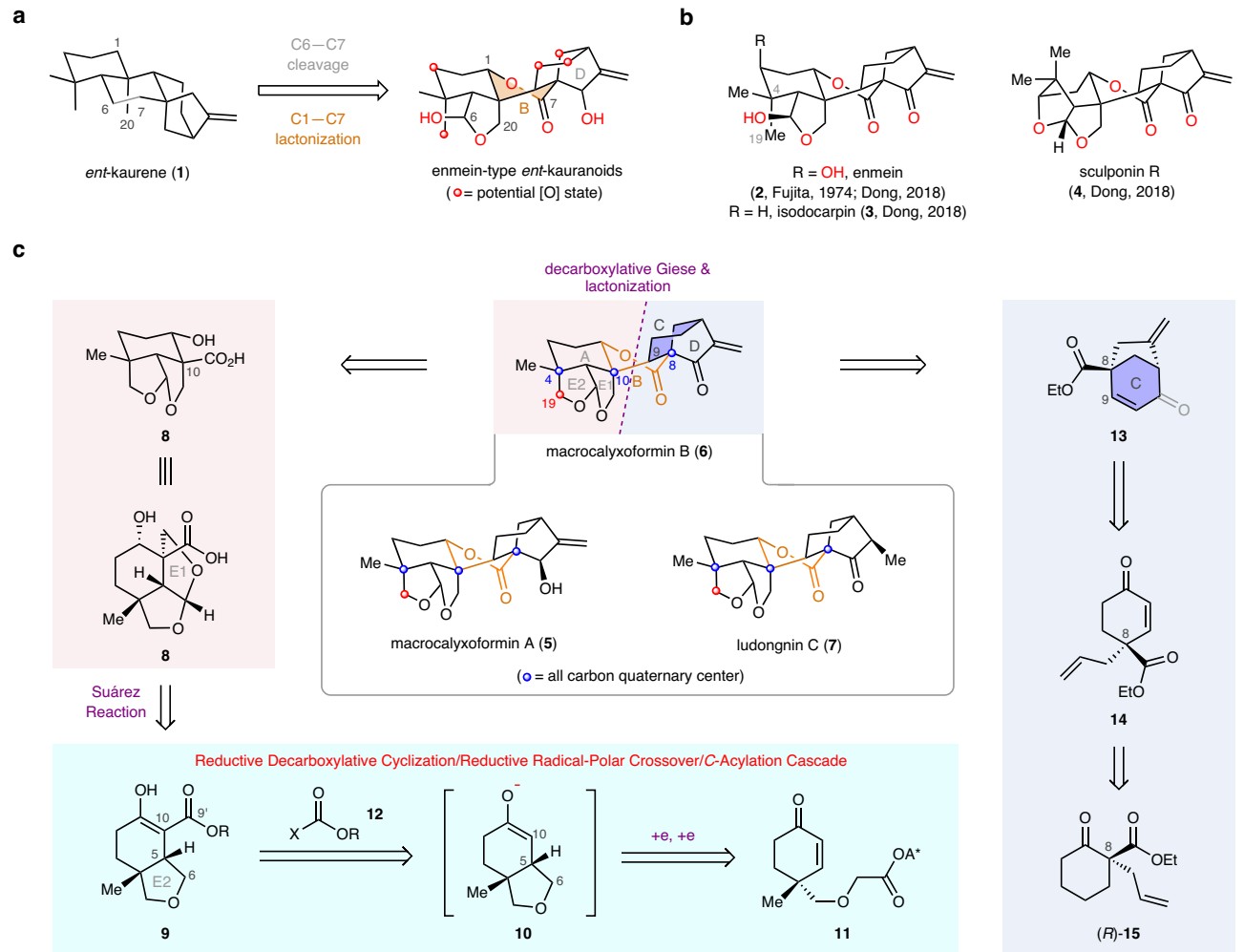

**Fig. 1 | Proposed biosynthesis and representative molecules of enmein-type *ent*-kauranoids and outlines of the synthetic plan. a** The biosynthetic pathway of enmein-type *ent*-kauranoids. **b** Successfully synthesized enmein-type *ent*-kauranoids. **c** Retrosynthetic analysis of C19-oxygenated enmein-type *ent*-kauranoids.

non-trivial challenges associated with constructing the twist-boat B lactone ring, we present herein our synthetic endeavors toward achieving their total synthesis.

## Results

### Synthetic Planning

We envisioned that macrocalyxoformin A (**5**) and ludongnin C (**7**) could be respectively accessed from macroclyxoformin B (**6**) via stereoselective 1,2 and 1,4-reductions of the enone moiety on the D ring (Fig. 1c). Following topological principles for retrosynthetic analysis[22], we prioritize disconnection of the most centrally located B lactone ring of **6** by cleaving the ester bond and the C9–C10 bond, the latter of which was anticipated to be forged via a decarboxylative Giese reaction[23–29] between the tertiary acid **8** and radicophile **13**. The E1 tetrahydrofuran ring of **8** was assumed to be accessible from **9** via an aldol reaction with formaldehyde followed by a Suárez modified Hofmann–Löffler–Freytag reaction[30]. To construct the β-keto ester **9**, we conceived an approach involving a reductive decarboxylative cyclization/reductive radical-polar crossover (RRPCO)[31–38]/C-acylation cascade, starting from readily available redox-active ester (RAE) **11**. In this process, the key carbon-carbon bond formations would result from an intermolecular radical conjugate addition (C5–C6 bond) and an acylation (C10–C9′ bond) reaction. Radicophile **13** could be obtained from **14** through a palladium-catalyzed oxidative cyclization[39]. Compound **14** could be readily traced back to the easily prepared compound (*R*)-**15**[40].

Although the proposed reductive decarboxylative cyclization/ RRPCO/C-acylation cascade holds conceptual efficiency, it presents at least two considerable challenges: (i) a viable acylation reagent with proper reactivity has to be selected: if the acylation reagent is not reactive enough, then the carbanion **10** might be acylated by the starting material RAE **11**; conversely, if the acylation reagent is too reactive, then could be reduced prior to the reduction of **11** or the radical precursor of **10**[41]. (ii) the desired C-acylation has to override the O-acylation[42]. Approaching these challenges, we thought that $CO_2$ would be a good choice[43–46]; however, considering the lability of the β-keto acid product, the additional esterification step, and the potential scalability issue in the early stage of the total synthesis, we prioritize the search for an appropriate acylation reagent.

### Preparation of the precursors and optimization of the reductive decarboxylative cyclization/RRPCO/C-acylation cascade

The preparation of the RAE **11** commenced with the alcohol (*R*)-**16** (Table 1), which could be easily prepared in 3 steps at decagram scale in 87% ee and 63% overall yield using Rawal's procedure[47]. Etherification of alcohol **16** with *tert*-butyl bromoacetate (NaOH, TBAB, quant.) provided *tert*-butyl ester **17**. Removal of the carboxyl *tert*-butyl protection (TFA) provided carboxylic acid, which can be converted to an array of RAEs in good-to-excellent yields.

We began the investigation of the proposed reductive decarboxylative cyclization/RRPCO/C-acylation cascade using the canonical ester **11a** as starting material (83% yield from *tert*-butyl ester **17**),

**Table 1 | Preparation of the RAEs and optimization of the decarboxylative cyclization/RRPCO/C-acylation**

| Entry[a] | 11 | 12 | [Ni] | Ligand | Temperature (°C) | 9 [yield (%)][b] |
|---|---|---|---|---|---|---|
| 1 | 11a | 12a | NiBr₂·DME | – | 50 | 9a (14%) |
| 2 | 11a | 12b | NiBr₂·DME | – | 50 | 9b (ND) |
| 3 | 11a | 12c | NiBr₂·DME | – | 50 | 9c (ND) |
| 4 | 11a | 12d | NiBr₂·DME | – | 50 | 9d (27%) |
| 5 | 11a | 12e | NiBr₂·DME | – | 50 | 9d (ND) |
| 6 | 11a | 12f | NiBr₂·DME | – | 50 | 9a (ND) |
| 7 | 11a | 12d | NiBr₂·DME | bpy | 50 | 9d (18%) |
| 8 | 11a | 12d | NiBr₂·DME | dtbbpy | 50 | 9d (21%) |
| 9 | 11a | 12d | NiCl₂·DME | – | 50 | 9d (25%) |
| 10 | 11a | 12d | Ni(COD)₂ | – | 50 | 9d (6%) |
| 11 | 11a | 12d | Ni(acac)₂ | – | 50 | 9d (3%) |
| 12 | 11a | 12d | NiBr₂·DME | – | RT | 9d (16%) |
| 13 | 11a | 12d | NiBr₂·DME | – | 70 | 9d (22%) |
| 14 | 11b | 12d | NiBr₂·DME | – | 50 | 9d (ND) |
| 15 | 11c | 12d | NiBr₂·DME | – | 50 | 9d (ND) |
| 16 | 11d | 12d | NiBr₂·DME | – | 50 | 9d (53%, 49%[c]) |

The numbers in bold in the table represent compound numbers. *TBAB* tetrabutylammonium bromide, *TFA* trifluoroacetic acid, *DIC* N,N'-diisopropylcarbodiimide, *DME* 1,2-dimethoxyethane, *DMAP* 4-dimethylaminopyridine, *Me* methyl, *iPr* isopropyl, *Bn* benzyl, *tBu* tert-butyl, *Ni(COD)₂* bis(cyclooctadiene)nickel(0), *Ni(acac)₂* nickel(II)bis(acetylacetonate), *NMP* N-methyl-2-pyrrolidone, *bpy* 2,2'-bipyridine, *dtbbpy* 4,4'-di-tert-butyl-2,2'-dipyridine, *Boc₂O* di-tert-butyl dicarbonate, *CbzCl* benzyl chloroformate, *e.e.* enantiomeric excess, *ND* not detected, *RT* room temperature.

[a]Reactions were conducted at a 0.15 mmol scale in 2 mL of NMP.

[b]Yields were determined by LC/MS.

[c]Isolated yield.

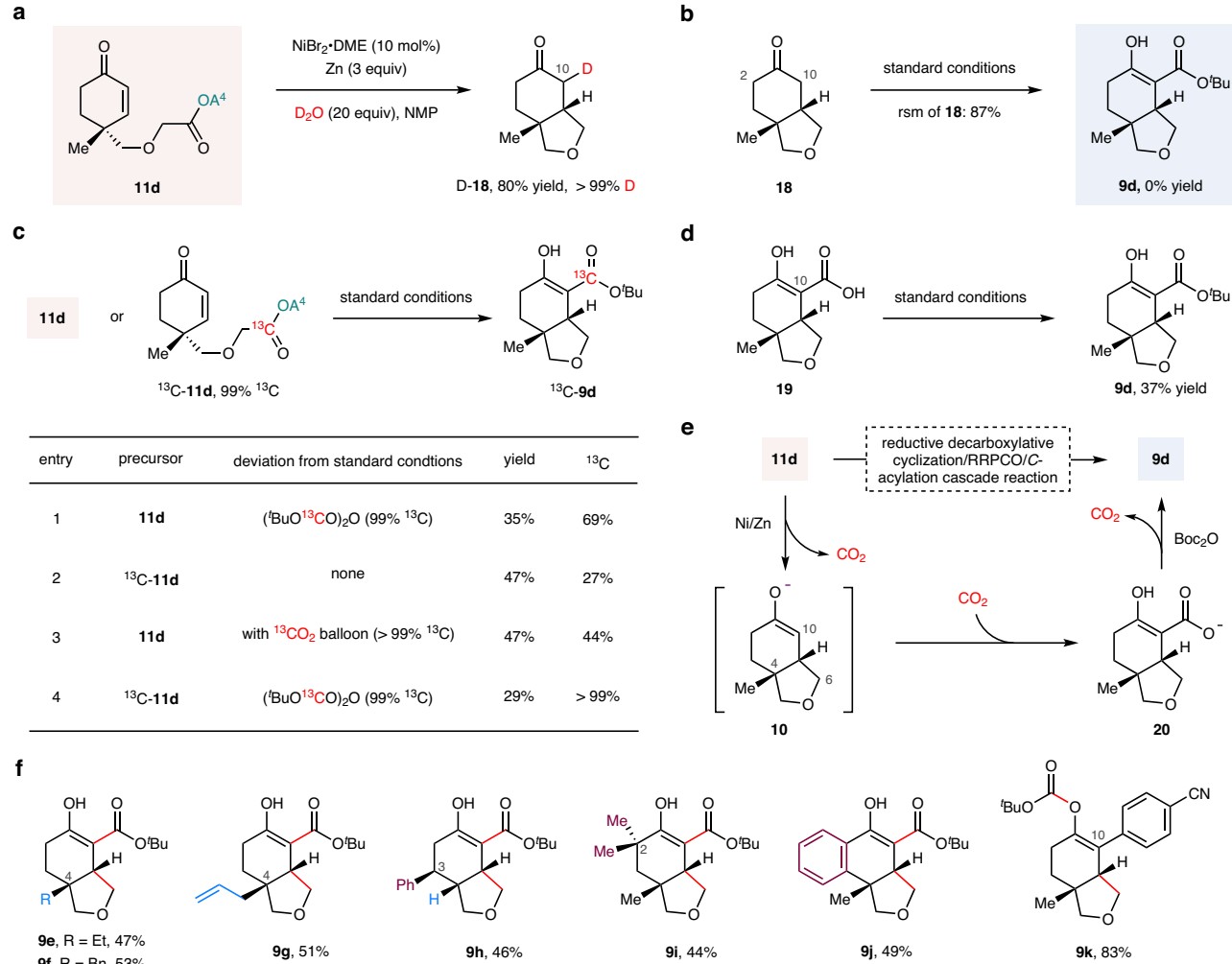

**Fig. 2 | Mechanistic studies and the proposed mechanism. a, b** Evidences for RRPCO. **c** Tracing the source of the C10 ester group. **d** Boc₂O was used in the esterification of carboxylic acid **19**. **e** Proposed reaction mechanism. **f** Substrate scope of the reductive decarboxylative cyclization/RRPCO/C-acylation cascade.

Abbreviations: DME, 1,2-dimethoxyethane; D₂O, deuterium oxide; NMP, *N*-mehtyl-2-pyrrolidone; A⁴, 1,8-naphthalimide; Boc₂O, di-*tert*-butyl decarbonate; Et, ethyl; Bn, benzyl; RRPCO, reductive radical-polar crossover; ¹³C, carbon-13; rsm, recovered starting material.

NiBr₂•DME as catalyst, and Zn as reductant. After extensive screening of potential acylation reagents (entries 1–6, Table 1), we were pleased to find that the desired product **9d** could be produced in 27% yield when di-*tert*-butyl decarbonate (Boc₂O) was employed, while other tested acylation reagents (**12a–f**) performed sluggishly. An extensive investigation of bipyridine ligands (entries 7 and 8), alternative catalysts (entries 9–11), and the reaction temperature (entries 12 and 13) were unfruitful. Eventually, we were happy to find that changing *N*-hydroxyphthalimide (NHPI) ester **11a** to *N*-(acyloxy)−1,8-naphthalimide **11d** increased the yield of **9d** to 53% (49% isolated yield, entry 16). Additionally, it is noteworthy that this cascade reaction can be performed at a decagram scale, easily producing multigram quantities of **9d** in one pot (42% isolated yield).

## Mechanistic studies

Although our proposed reductive decarboxylative cyclization/RRPCO/C-acylation cascade showed good efficiency in the synthesis of the bicyclic β-keto ester **9d**, the use of Boc₂O as the *C*-acylation reagent seems counterintuitive because it is typically used to introduce the Boc protecting group to amine functionalities, with only a few examples as a *C*-acylation reagent reported to date[48–50]. We, therefore, performed a series of experiments to gain further insight into the mechanism of this cascade reaction (Fig. 2).

The reductive radical cyclization/RRPCO sequence was inferred from our findings that (i) replacing Boc₂O (6 equiv) with D₂O (20 equiv) in the standard reaction conditions provided the only observed product **18** in 80% yield with >99% deuterium incorporation (Fig. 2a), indicating a sequence of decarboxylative 5-*exo*-trig radical cyclization and an RRPCO of C10 radical occurred; (ii) no **9d** was detected when **18** was applied as the starting material instead of RAE **11d** (Fig. 2b), excluding the possibility that **18** was the precursor of the acylation. Moreover, we found that Boc₂O was not a good *C*-acylation reagent for **18**, as treatment of **18** with a variety of bases (*e.g.* LDA, NaHMDS) followed by adding Boc₂O only provided trace amounts of **9d** and the C2 acylated isomer (Supplementary Fig. 6).

We next focused on identifying the source of the C10 ester group of **9d** (Fig. 2c) via a series of isotope labeling experiments. Initially, ¹³C labeled Boc₂O was prepared [¹³CO₂ (>99% ¹³C), *t*BuOK, MsCl, pyridine, Supplementary Fig. 7] and used instead of Boc₂O. We obtained **9d** with 69% ¹³C incorporation (35% yield, Fig. 2c, entry 1), which indicated that the C10 ester group was not fully derived from Boc₂O. Consequently, we labeled the RAE group of **11d** with ¹³C (see Supplementary Figs. 11 and 12 for its preparation) and subjected ¹³C-**11d** to the standard reaction conditions, we detected the product **9d** with 27% ¹³C incorporation (47% yield, Fig. 2c, entry 2), which indicated that the C10 ester group is partially derived from the reaction of the C10 carbanion with

CO₂ released by the decarboxylation process. To further demonstrate this point, we performed the standard reaction under a $^{13}CO_2$ (>99% $^{13}C$) atmosphere (Fig. 2c, entry 3), and a 44% $^{13}C$ incorporation of product **9d** was observed.

On the basis of these lines of evidence, a putative mechanism for the acylation process is proposed (Fig. 2e). The carbanion **10** (was generated through the reductive decarboxylative cyclization/RRPCO process) reacts with $CO_2$ produced by the decarboxylation, giving rise to the carboxylate **20**. Esterification of **20** by Boc₂O affords the desired β-keto esters **9d** along with one $^tBuO^-$ and two $CO_2$[51]. The released $CO_2$ could also be involved in the carboxylation of carbanion **10**, accounting for the high efficiency of the acylation and the relatively low $^{13}C$ incorporation in the above isotope labeling experiments. Note that: (i) we detected >99% $^{13}C$ incorporation of **9d** when Boc₂O and RAE **11d** were replaced by $^{13}C$ labeled Boc₂O and $^{13}C$-**11d** simultaneously in standard reaction conditions (Fig. 2c, entry 4), (ii) Additionally, we found that treatment of the carboxylic acid **19** with our standard conditions (Fig. 2d) could afford the esterification product **9d** in 37% yield. Both experiments support the plausibility of our proposed reaction mechanism.

Armed with a comprehensive understanding of the mechanism and optimal conditions, we next briefly explored the generality of this cascade process (Fig. 2f). Intriguingly, displacing the methyl group on the existing C4 quaternary carbon to an ethyl group (**9e**), a benzyl group (**9f**), an allyl group (**9g**), or removal of the C4 methyl group (**9h**) did not compromise the yield. Furthermore, the substrates with substitutions at C3 and C2 were also amenable to this cascade reaction, delivering **9h**–**9j** in synthetically useful yields. Notably, the incorporation of an aromatic ring at the α-position of the α, β-unsaturated ketones (C10) resulted in the formation of the *O*-acylated product **9k** in high yield, rather than the expected *C*-acylated product, indicating a significant impact of steric hindrance on the *C*-acylation.

## Building the E1 ring and the radicophile 13: challenges in decarboxylative Giese reaction

With a reliable and scalable synthesis of **9d** in hand, we set out to construct the E1 THF ring (Fig. 3a). Treatment of **9d** with formalin in the presence of Yb(OTf)₃ gave rise to the C10 aldol product **21** in a completely stereoselective manner[52]. The high stereoselectivity is plausibly attributable to the influence of the axial methyl group at C4[20]. Subjection of **21** to the conditions reported by Suárez and co-workers (PIDA, I₂, hν) forged the E1 THF ring smoothly[30], producing **22** in good yield (61% from **21**). Notably, a ring flip process occurred during the reaction. Reduction of the C1 ketone of **22** with a sterically hindered reducing reagent LiAl(O$^t$Bu)₃H afforded the desired alcohol **23** with excellent diastereoselectivity (9:1, 80%), while other less hindered reagents (e.g., NaBH₄, DIBAL-H) proved to be unviable. Having secured access to **23**, we rapidly prepared two precursors (**24** and **25**), which can be used to construct the C9–C10 bond via the decarboxylative Giese reaction: deprotection of the $^t$Bu ester of **23** with trifluoroacetic acid (TFA), protection of the C1 alcohol and carboxylic acid with *tert*-butyldimethylsilyl (TBS) group and hydrolysis of TBS ester produced acid **24** (93%), which was sequentially activated with NHPI to afford the RAE **25** (66%).

Preparation of radicophile **13** began with an asymmetric α-allylation of β-ketoester **26**[40], giving (*R*)-**15** in 96% yield with 96% e.e. (Fig. 3b). Treatment of (*R*)-**15** with NaBH₄, followed by mesylation and elimination of the resulting mesyl ester, produced alkene **27**, which underwent an allylic oxidation (CrO₃, TBHP), affording enone **14** (27% over 4 steps). Exposure of **14** with *tert*-butyldimethylsilyl trifluoromethanesulfonate (TBSOTf) in the presence of Et₃N and subjection of the resulting silyl enol ether with Pd(OAc)₂ furnished radicophile **13** (66% over 2 steps)[39], whose absolute configuration was confirmed by X-ray crystallography of its acid derivative **28**.

In the subsequent assembly process utilizing decarboxylative Giese reaction, despite our diligent efforts, employing the radicophile enone **13** in conjunction with various photoredox decarboxylation conditions using acid **24** (see Supplementary Fig. 31 for details)[23,24] or different reductive decarboxylation conditions using RAE **25** (see Supplementary Fig. 32 for details)[25–27] only resulted in direct decarboxylation rather than the desired coupling product **29** (Fig. 3c). We initially ascribed this outcome to the significant steric hindrance arising from the γ-quaternary carbon of α, β-unsaturated **13**. Notably, to our knowledge, no prior reports exist on the decarboxylative Giese reaction of tertiary radicals with α, β-unsaturated acceptors containing a γ-quaternary carbon.

In an effort to mitigate the steric hindrance associated with the radicophile, we chose to employ α, β-unsaturated lactone (*S*)-**31**[53] (Fig. 3d, see Supplementary Fig. 30 for its enantioselective preparation), which lacks substitution on the carbon adjacent to the reacting carbon centers of the radicophile. Notably, this approach could capitalize on the intrinsic configurations of C13 in (*S*)-**31**, potentially leading to the desired stereochemistry at C9. Subsequent B lactone ring construction could employ a lactonic Ireland–Claisen rearrangement[54]. However, none of the decarboxylative conditions utilizing acid **24** and RAE **25** yielded the desired coupling product with (*S*)-**31**, underscoring the pivotal role of steric effects caused by the substituents at the reacting carbon centers of the α, β-unsaturated ester.

We also investigated the intramolecular Giese reaction using acids **34**–**36** (Fig. 3e, see Supplementary Figs. 34–36 for their preparations) and the RAEs thereof **37**–**39** (see Supplementary Fig. 38 for their preparations); unfortunately, we could not detect the desired coupling products. We speculated that the failures were due to (i) the C1-ester groups not favoring the *cis* configuration needed in the reaction transition state and (ii) the relatively high barrier of rotation around the ester C–O bond[55] to the desired reaction transition state. Furthermore, attempts to use the C10-acyl telluride as the tertiary radical precursor[56] also proved unsuccessful in both intermolecular and intramolecular Giese reactions (see Supplementary Figs. 33 and 40 for details).

## Completion of the total synthesis

Recognizing the pivotal role of the steric effects of the radicophile in constructing the C9–C10 bond, we chose to use the simplest acyclic unsaturated esters with no substitution on the reacting carbon centers[20,57,58]. As shown in Fig. 4, the protection of the C1–OH of **23** with a benzyl group (NaH, BnBr, 84%) followed by deprotection of the C10-*tert*-butyl ester (TFA) and activation of the resulting acid with NHPI (DIC, DMAP) provided RAE **40** in an excellent yield (99%). The benzyl-protected RAE **40** was not used in the investigate of the aforementioned intermolecular decarboxylative Giese reaction (Fig. 3), due to the presence of potential coupling products containing alkene group, which would be incompatible with the benzyl deprotection. Subjection of RAE **40** with 2,2,2-trifluoroethyl acrylate under Baran's decarboxylative Giese reaction conditions [Ni(ClO₄)₂·6H₂O, Zn, LiCl] successfully afforded the desired coupling product **41** (75% NMR yield)[26]. Subsequent removal of the benzyl group under hydrogenation conditions (H₂, Pd/C), followed by a spontaneous lactonization, yielded lactone **42**, whose structure was confirmed by X-ray crystallography.

Oxidative dehydrogenation of lactone **42** [(PhSeO)₂O, 49% from **40**][59], followed by a conjugate addition of allyl cuprate species (allylMgBr, CuBr·DMS, LiBr) to the resulting α, β-unsaturated lactone afforded **43** (96%) with complete diastereoselectivity. Note that the conformation of the lactone ring was converted from a twist-boat to a half-chair conformation in this process. Deprotonation of lactone **43** (LDA, HMPA), followed by allylation with 2,3-dibromopropene produced **44** (43%, 57% brsm, d.r. = 10:1). Interestingly, the conformation of the lactone ring was transformed back to a twist-boat, and the newly installed allyl group occupied an equatorial position. The

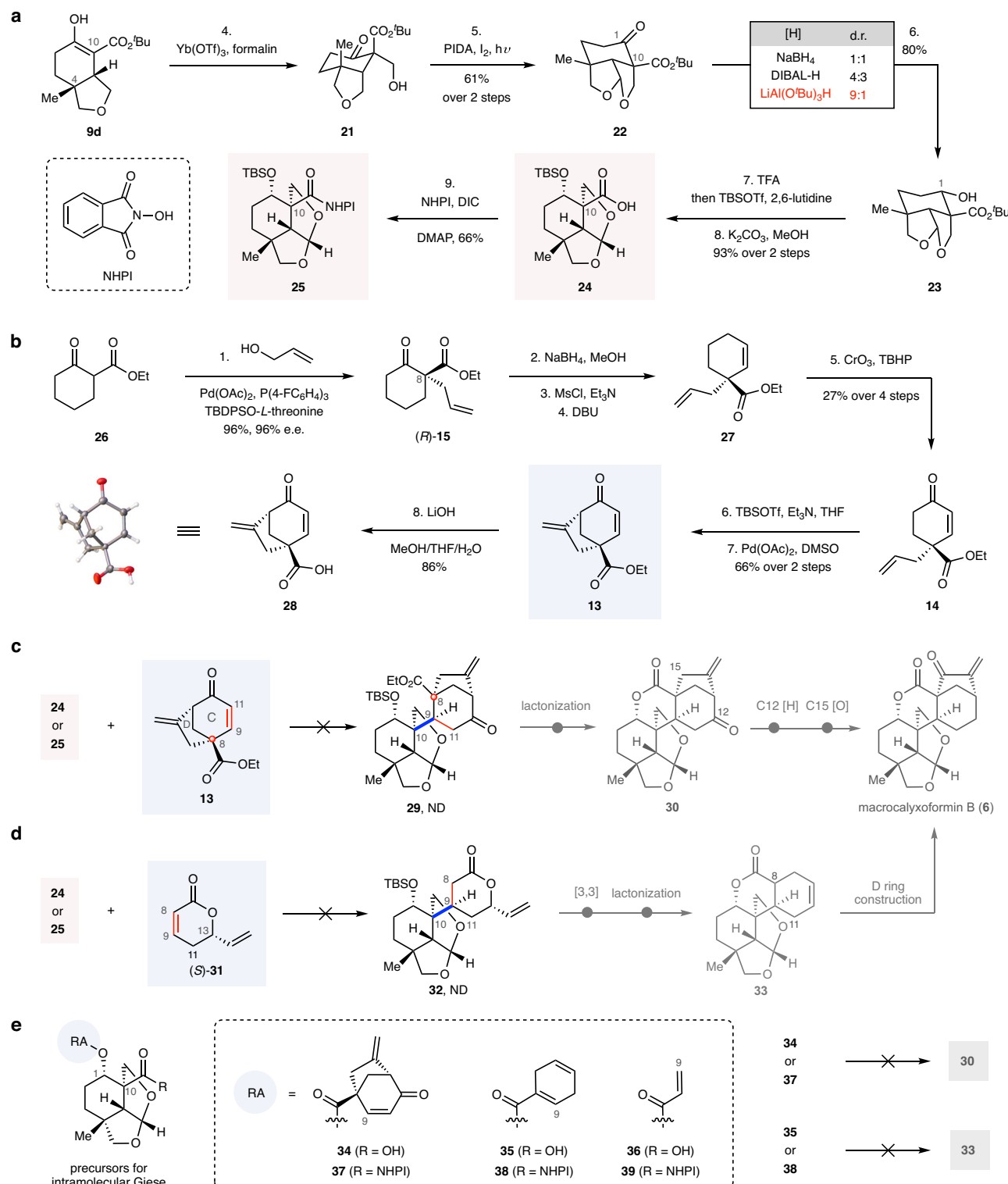

**Fig. 3 | Preparation of precursors and attempts for the decarboxylative Giese reaction. a** Stereoselective construction of the E1 ring and the C1−OH. **b** Asymmetric synthesis of radicophile **13**. **c** Attempts with audiophile enone **13**. **d** Attempts with radicophile (*S*)−**31**. **e** Attempts at the decarboxylative intramolecular Giese reaction. *Abbreviations*: *t*Bu *tert*-butyl, PIDA phenyliodine diacetate, hv light irradiation, DIBAL-H diisobutylaluminium hydride, TFA trifluoroacetic acid, TBSOTf *tert*-butyldimethylsilyl trifluoromethanesulfonate,

NHPI *N*-hydroxyphthalimide, DIC *N,N*′-diisopropylcarbodiimide, DMAP 4-dime-thylaminopyridine, Ac acetyl, TBDPS *tert*-butyldiphenylsilyl, MsCl methane-sulfonyl chloride, DBU 1,8-diazabicyclo[5,4,0]undec-7-ene, TBHP *tert*-butyl hydroperoxide, THF tetrahydrofuran, DMSO dimethyl sulfoxide, d.r. diaster-eomer ratio, e.e. enantiomeric excess, [H] hydrogenation, [O] oxidation, [3,3] 3,3-sigmatropic rearrangement, ND not detected, RA radical acceptor.

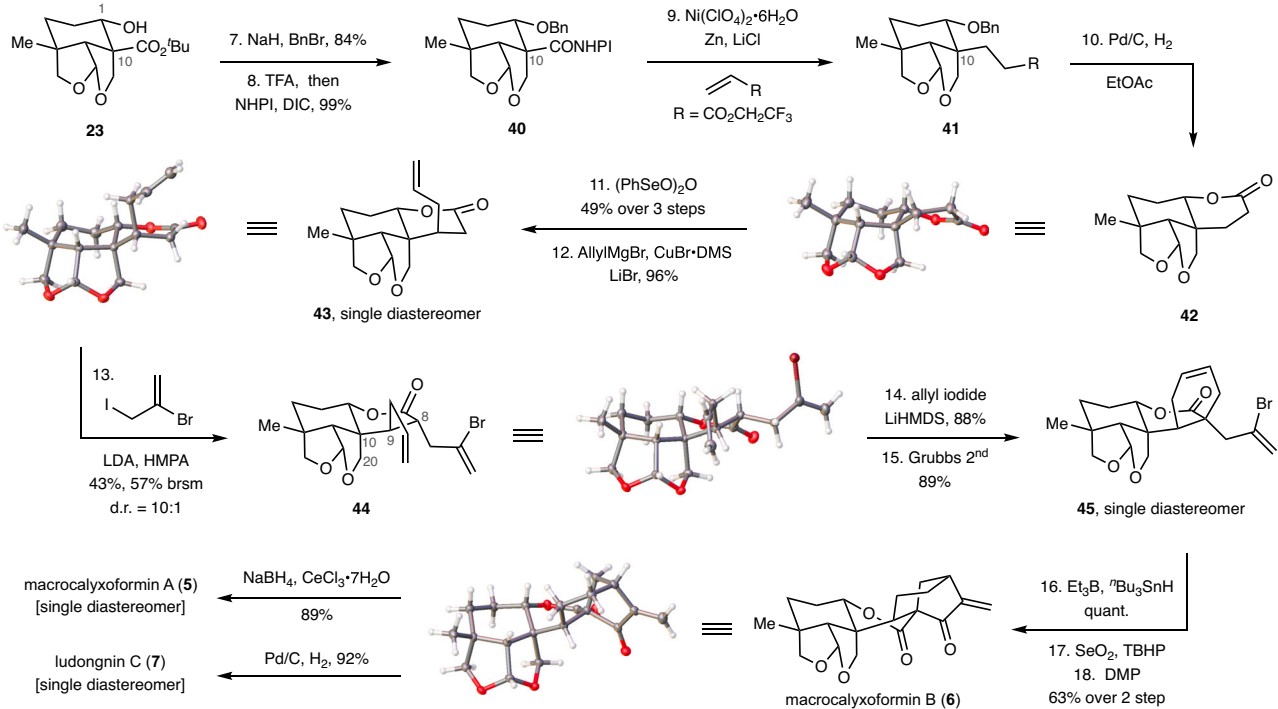

**Fig. 4 | Total syntheses of (−)-macrocalyxoformins A and B and (−)-Ludongnin C.** All yields were determined by isolation. *Abbreviations*: BnBr benzyl bromide, TFA trifluoroacetic acid, NHPI *N*-hydroxyphthalimide, DIC *N,N'*-diisopropylcarbodiimide, DMS dimethyl sulfide, LDA lithium diisopropylamide, HMPA hexamethylphosphoramide, LiHMDS lithium bis(trimethylsilyl)amide, *n*Bu *n*-butyl, TBHP *tert*-butyl hydroperoxide, DMP Dess−Martin periodinane, brsm based on recovered starting material, d.r. diastereomer ratio.

stereoselectivity of this allylation and ring flip process is presumably caused by the steric repulsion of the vicinal C9−allyl group and the 1,3-diaxial effect of the C10−C20 bond.

Allylation of the lactone ring of **44** (LiHMDS, allyl iodide, 88%) followed by ring-closing metathesis (RCM) reaction of the resulting triene using Grubbs 2nd catalyst proceeded with excellent stereo- and chemoselectivity, giving rise to **45** in an excellent yield (89%). It is noteworthy that (i) the 1,3-diaxial effect caused by the C10−C20 bond probably overrode the steric influence exerted by the vicinal C9−allyl group, resulting in the superb stereoselectivity of this allylation; (ii) the vinyl bromide moiety was not disturbed during the RCM process[60]. Finally, the D ring of macroclyxoformin B (**6**) was smoothly constructed from **45** via a classic sequence involving a 5-*exo* radical annulation (Et$_3$B, $^n$Bu$_3$SnH, quant.)[17,61], an allylic oxidation (SeO$_2$, TBHP), and a Dess-Martin oxidation (63% over 2 steps). The structure of macroclyxoformin B (**6**) was confirmed by X-ray crystallography. Consequently, respective Luche reduction (NaBH$_4$, CeCl$_3$·7H$_2$O) and

hydrogenation (H$_2$, Pd/C) of macrocalyxoformin B (**6**) produced macrocalyxoformin A (**5**, 89%) and ludongnin C (**7**, 92%), both as single diastereomers.

**Anticancer activity evaluation of the synthetic natural products**

The α-methylenecyclopentanone system (D ring) in *ent*-kauranoids is recognized as a crucial pharmacophore for their antitumour activity[1]. Our synthesized natural products **5**–**7** exemplify this role effectively. We evaluated their impact on cell viability across nine cancer cell lines from five human tissues. Macrocalyxformin B (**6**) displayed significant broad-spectrum anticancer activity at the micromolar level, as shown in Table 2. Conversely, macrocalyxformin A (**5**) and ludongnin C (**7**) exhibited negligible and weak activity, respectively, across the tested cancer cell lines, reaffirming the essential role of the α-methylenecyclopentanone moiety in anti-tumor activity. Notably, the comparatively stronger anti-tumor activity observed in compound **7**, in contrast to compound **5**, implies a potential non-covalent interaction facilitated by the ketone group, thus highlighting a promising direction for future exploration.

## Discussion

In summary, we have achieved the enantioselective total syntheses of three C19-oxygenated enmein-type *ent*-kauranoids: (−)-macro-calyxoformins A (**5**) and B (**6**) and (−)-ludongnin C (**7**). The enabling basis for this total synthesis is a devised Ni-catalyzed reductive decarboxylative cyclization/RRPCO/*C*-acylation cascade that allowed efficient construction of the E2 THF ring and an adjacent β-keto ester group, which served as a handle for the installation of the E1 THF ring and the B lactone ring. Our mechanistic investigations revealed that the acylation step in this cascade is realized by a carboxylation of carbanion followed by an in situ esterification. We anticipate that the reductive decarboxylative cyclization/RRPCO/*C*-acylation cascade reaction we developed could be extended to the syntheses of other highly oxidized and polycyclic natural products. Evolutionary studies on radicophiles for decarboxylative

**Table 2 | IC$_{50}$ values (µM) of synthetic natural products against 9 different cancer cell lines**

| Cell lines | Cancer type | 5 | 6 | 7 |
|---|---|---|---|---|
| ME-180 | Cervical cancer | N/A | 2.21 | 31.08 |
| U2OS | Osteosarcoma | N/A | 3.09 | >100 |
| A549 | Nonsmall cell lung cancer | N/A | 4.49 | 65.0 |
| HCT116 | Colorectal cancer | N/A | 1.34 | 24.24 |
| SW756 | Cervical cancer | N/A | 2.07 | 14.95 |
| HeLa | Cervical cancer | N/A | 2.48 | 30.88 |
| SiHa | Cervical cancer | N/A | 1.80 | 20.05 |
| HuH-7 | Hepatocellular carcinoma | N/A | 3.95 | >100 |
| SK-CO-1 | Colorectal cancer | N/A | 3.97 | 38.03 |

The numbers in bold in the table represent compound numbers.

Giese reaction, aimed at constructing the C10 quaternary carbon, revealed the challenges posed by radicophiles with substituents at the reacting carbon centers. Further anti-tumor examination of these synthetic natural products highlighted the crucial role of the ketone on the D ring.

## Methods

### General procedure for the preparation of redox-active esters

To a cooled (0 °C) solution of the *tert*-butyl ester (1.0 equiv) in $CH_2Cl_2$ (0.3 M) was added TFA (0.6 mL/mmol *tert*-butyl ester). The reaction mixture was warmed to room temperature and stirred for 3 h. The reaction mixture was concentrated directly, giving the carboxylic acid, which was used directly for the next step without further purification. Note: To rapidly remove TFA completely, the above crude product can be dissolved in toluene and concentrated under vacuum; this process can be repeated until no TFA can be detected by $^{19}F$ NMR. The *N, N'*-diisopropylcarbodiimide (DIC, 1.2 equiv) was added dropwise to a cooled (0 °C) mixture of above carboxylic acid (1.0 equiv), AOH (NHPI or its analogs, 1.1 equiv), and 4-dimethylaminopyridine (DMAP, 0.3 equiv) in anhydrous $CH_2Cl_2$ (0.2 M). After 4 h stirring at room temperature, the reaction mixture was directly concentrated under reduced pressure. Purification by flash column chromatography (silica gel) gave the RAE.

### General procedure for the reductive decarboxylative cyclization/radical-polar crossover/*C*-acylation cascade

In a glovebox, Boc₂O (6.0 equiv) was added to the mixture of the RAE (1.0 equiv), NiBr₂·DME (10 mol%), and Zn (3.0 equiv) in anhydrous *N*-mehtyl-2-pyrrolidone (NMP, 0.075 M) at room temperature. The reaction mixture was then moved out of the glove box and heated to 50 °C. After stirring for 14 h the reaction mixture was filtered through a pad of Celite®, and the filter was washed with EtOAc. The filtrate was washed with $H_2O$ (2 times), brine (1 time), whereby the aqueous layers were back-extracted with EtOAc (3 times). The combined organic layers were dried over $Na_2SO_4$, filtered and concentrated under reduced pressure to give the crude product. Purification by column chromatography (silica gel) gave the product.

### Cell culture

The human cell lines ME-180, U2OS, A549, HCT-116, SW756, HeLa, SiHa, HuH-7, and SK-CO-1 were obtained from Cell Resource Center, Peking Union Medical College (Beijing, China). All cell lines were confirmed to be mycoplasma-free by PCR. Regular adherent cell culture methods were used to culture cells in tissue-culture incubators with 5% $CO_2$ at 37 °C. A549 was grown in RPMI-1640 medium with 10% fetal bovine serum (FBS) and 2 mM L-glutamine. SK-CO-1 was grown in MEM medium with 10% FBS and 2 mM L-glutamine. All other cells were grown in DMEM medium with 10% FBS and 2 mM L-glutamine.

### Cell viability assay

Three thousand cells in 100 μL of medium were plated per well in 96-well flat clear bottom white polystyrene TC-treated microplates (Corning, USA). Then cells were dosed with a serial dilution of compounds with a D300e digital dispenser (Tecan, Männedorf, Switzerland). Cell survival was measured 72 h later using CellTiter-Glo luminescent cell viability assay kit (Promega, Madison, USA) according to the manufacturer's instructions. Luminescence was recorded by EnVison multimode plate reader (PerkinElmer, Waltham, USA). $IC_{50}$ was determined with GraphPad Prism v8.0.2 using baseline correction (by normalizing to DMSO control), the asymmetric (four parameters) equation, and the least-squares fit.

### Reporting summary

Further information on research design is available in the Nature Portfolio Reporting Summary linked to this article.

## Data availability

The X-ray crystallographic coordinates for structures reported in this study have been deposited at the Cambridge Crystallographic Data Centre (CCDC) under deposition numbers 2238607 (**28**), 2238608 (**42**), 2238609 (**43**), 2238610 (**44**), 2238611 (**6**). Copies of the data can be obtained free of charge via https://www.ccdc.cam.ac.uk/structures/. All other data supporting the findings of this study, including experimental procedures and compound characterization, NMR, and HPLC, are available within the Article and its Supplementary Information and all data are available from the corresponding author upon request.

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

## Acknowledgements

Financial support for this work was provided by the National Natural Science Foundation of China (Grant No. 22171025 to C.L.), MOST of China (to C.L.), and Tsinghua Institute of Multidisciplinary Biomedical Research, Tsinghua University (to C.L.). We thank Masayuki Inoue (University of Tokyo) and Mingji Dai (Emory University) for the helpful discussion.

## Author contributions

Z.C., W.S., S.Y., and C.L. conceived and designed the experiments. C.L. directed the project. Z.C., W.S., Jingfu Z., and Junming Z. carried out the experiments. Z.C., W.S., and C.L. interpreted the results. Y.M. and X.S. performed the HRMS data collection and analysis. T.H. and P.L. evaluated the $IC_{50}$ values of natural products. C.L. and Z.C. wrote the manuscript with input from all other authors.

## Competing interests

The authors declare no competing interests.
