## [Peer Review File · Nature Communications]

Total Syntheses of (–)-Macrocalyxoformins A and B and (–)-Ludongnin CREVIEWER COMMENTS

Reviewer #1 (Remarks to the Author):

Recommendation: Publish in Nature Communications after minor revisions.

Comments:

This is an excellent synthesis from Prof. Chao Li and co-workers. This paper reports a reductive decarboxylative cyclization/radical-polar crossover/C-acylation cascade reaction that enables access to the substructure of C-19 oxygenated enmein-type ent-kauranoids. This is a creative merger of previously known reactivity patterns with these systems and more contemporary reductive radical cascade process. As a practical application, the authors accomplished the first total synthesis of (-)- macrocalyxofornins A and B and (-)- ludongnin C. The main text and the experimental protocol are clear. Overall, the reviewer is very happy to recommend this manuscript to be published in Nature Communications if the authors could answer some minor revisions properly.

(1) The number of ¹H signals should match with the number of hydrogens present in the corresponding compounds; e.g., for compound rac-16 the total number of ¹H signals presented is 14, but the compound contains 12 hydrogen atoms. The same problem was found with compounds S14, S34 and 36.

(2) The HRMS mass does not match the reported formula for compounds S46 and S47.

(3) In Figure 2, I am interested in understanding whether a reaction can proceed smoothly when an alkyl group is assembled at the C-6 of substrate 11d.

Reviewer #2 (Remarks to the Author):

The authors describe the synthesis of the macrocalyxofornins A and B and ludongnin C. These natural products are characterized by a complex carbon framework with high density of chiral centers, oxygen functionalities and two chiral quaternary centers. Such natural products are investigated nowadays and the aim is to provide rapid access with eventually new strategies or new reactions. In this context the current paper is timely and appropriate for publication in nature communications. The authors present an elegant retrosynthetic analysis for their synthesis and propose a reductive radical-polar crossover C-acylation cascade initiated by decarboxylative cyclization. They present an optimization of this sequence together with a mechanistic investigation. These will help other research groups for implementing the cascade in other natural products syntheses.

in the course of their synthesis the encounter, not unexpected, problems joining the two pivotal segments (24 or 25) and 13. As they rationalize, this is probably due to steric hindrance unfolded by both segments. To circumvent the problem, they revise the synthesis and continue with a simple olefine (step 9, Fig. 4). This then leads to the targeted natural products.

At this point I have to remarks. However, these should not deter the paper from publication. It would be nice if the authors could comment, if they have tried to perform the pivotal coupling intramolecularly. Also, I would re-draw the Figure 1 as the successful synthesis is different from what was planned initially. The paper should not reflect chronological how the research was preformed but how the final synthesis looks like. In Figure 1 the reader is put

on a track which was not successful.

In the experimental, the α -D-value should also be provided for the intermediates.

Otherwise, I enjoyed reading the paper and recommend publication.

Reviewer #3 (Remarks to the Author):

The author reported the first and enantioselective total syntheses of C-19 oxygenated enmein-type ent-kauranoids, including (–)- macrocalyxofornins A and B, and (–)-ludongnin C. Key feature of the synthesis is a newly devised Ni-catalyzed reductive decarboxylative cyclization/radical-polar crossover/C-acylation cascade that forges a THF ring concomitantly with the β - keto ester group. A notable aspect of this synthesis is the innovative Ni-catalyzed reductive decarboxylative cyclization/radical-polar crossover/C-acylation cascade, which enables the formation of a THF ring concurrently with the β -keto ester group. Detailed investigation into the reaction mechanism of this cascade reveals that the C-acylation process involves carboxylation followed by in situ esterification. The manuscript is well written and the literature references are properly cited, and this reviewer supports the acceptance of this manuscript after minor revisions.

A few suggestions are listed as follows:

1. The isolated yield of the decarboxylative cyclization/RRPCO/C-acylation cascade of 11d is moderate. It would be helpful to know if any byproducts were identified and characterized.

2 . Regarding the proposed mechanism for the decarboxylative cyclization/RRPCO/C-acylation cascade (Supplementary Fig.14), the authors should comment on the possibility of a transesterification reaction in addition to the decarboxylation process involving the carbanion 20 reacting with Boc₂O to form intermediate III.

3. In the supporting information, the authors attempted to build up a molecular skeleton through an intramolecular free radical cyclization. Although the target product was not obtained, the authors also mentioned the intermolecular approach. It is suggested that the authors summarize and reflect these outcomes in the main text together to provide a complete story.

4. The authors mentioned a good yield in the free radical cyclization reaction of substrate 38. It is curious whether oxygen and other initiators need to be added during the operation, as the reaction is carried out in an argon atmosphere according to the supporting information.

5. Page S40, regarding preparation of substrate S33, does the additional iodine in the system require reductive quenching? Is there anything worth noting?

7. Both the main text and supporting information require further editorial modifications. The third paragraph, page 10: “an” should be deleted, and “lactones” should be changed into “lactone”.

The first paragraph, page 11: “giving raise to” should be changed into “giving rise to”, and “override” should be changed into “override”.

The third paragraph, page S48: “provide” should be changed into “was provided”.

Page S49: the mass spectrometry data of product 25 needs to be left aligned.

Point-by-Point Response to Reviewers Comments

Reviewer #1 (Remarks to the Author):

Recommendation: Publish in Nature Communications after minor revisions.

Comments:

This is an excellent synthesis from Prof. Chao Li and co-workers. This paper reports a reductive decarboxylative cyclization/radical-polar crossover/*C*-acylation cascade reaction that enables access to the substructure of C-19 oxygenated enmein-type *ent*-kauranoids. This is a creative merger of previously known reactivity patterns with these systems and more contemporary reductive radical cascade process. As a practical application, the authors accomplished the first total synthesis of (–)-macrocalyxoformins A and B and (–)-ludongnin C. The main text and the experimental protocol are clear. Overall, the reviewer is very happy to recommend this manuscript to be published in Nature Communications if the authors could answer some minor revisions properly.

Response: Let us express our gratitude to the Referee for the encouraging words and the insightful suggestions provided below. We have addressed each of the corrections/comments as follows:

(1) The number of ¹H signals should match with the number of hydrogens present in the corresponding compounds; e.g., for compound *rac*-16 the total number of ¹H signals presented is 14, but the compound contains 12 hydrogen atoms. The same problem was found with compounds **S14**, **S34** and **36**.

Response: Thank you for pointing out these discrepancies. We sincerely apologize for the errors in the previously reported ¹H NMR data. To clarify: i) The NMR spectra for the mentioned compounds [*rac*-16, **S14**, **S34**, and **36** (*N.B.* current compound **43**)] are correct; the errors were due to typographical mistakes. ii) We have carefully rechecked and verified all NMR data in the Supplementary Information to ensure their accuracy. Thank you for your understanding.

(2) The HRMS mass does not match the reported formula for compounds **S46** and **S47**.

Response: We appreciate the Referee for pointing out these mistakes. Similar to the situation mentioned above, the HRMS data was correct, but the molecular formulas were incorrect due to

typographical errors. We have also carefully rechecked and verified all HRMS data in the Supplementary Information to ensure their accuracy.

(3) In Figure 2, I am interested in understanding whether a reaction can proceed smoothly when an alkyl group is assembled at the C-6 of substrate **11d**.

Response: We are grateful for the Referee's direction regarding the impact of the alkyl substitution at C-6 of **11d**. Following this suggestion, we prepared the substrate **S37** with a methyl group at C-6 to examine its viability in our key reaction. Unfortunately, the desired product **S44** was obtained in only 8% yield. The low yield may be attributed to the steric repulsion between the C-4 quaternary carbon and the C-6 secondary radical. Therefore, it is challenging to form a polysubstituted and crowded tetrahydrofuran ring system under these conditions. The procedures for the preparation of **S37** and **S44**, along with the corresponding characterization data, have been included in the revised Supplementary Information.

Finally, let us again thank the Referee for the above exceptionally helpful suggestions.

Reviewer #2 (Remarks to the Author):

The authors describe the synthesis of the macrocalyxoformins A and B and ludongnin C. These natural products are characterized by a complex carbon framework with high density of chiral centers, oxygen functionalities and two chiral quaternary centers. Such natural products are investigated nowadays and the aim is to provide rapid access with eventually new strategies or new reactions. In this context the current paper is timely and appropriate for publication in nature communications. The authors present an elegant retrosynthetic analysis for their synthesis and propose a reductive radical-polar crossover C-acylation cascade initiated by decarboxylative cyclization. They present an optimization of this sequence together with a mechanistic investigation. These will help other research groups for implementing the cascade in other natural products syntheses.

In the course of their synthesis the encounter, not unexpected, problems joining the two pivotal segments (**24** or **25**) and **13**. As they rationalize, this is probably due to steric hindrance unfolded by both segments. To circumvent the problem, they revise the synthesis and continue with a simple olefine (step 9, Fig. 4). This then leads to the targeted natural products.

At this point I have to remarks. However, these should not deter the paper from publication. It would be nice if the authors could comment, if they have tried to perform the pivotal coupling intramolecularly. Also, I would re-draw the Figure 1 as the successful synthesis is different from what was planned initially. The paper should not reflect chronological how the research was preformed but how the final synthesis looks like. In Figure 1 the reader is put on a track which was not successful.

In the experimental, the alpha-D-value should also be provided for the intermediates.

Otherwise, I enjoyed reading the paper and recommend publication.

Response: We would like to first express our gratitude to the Referee for the encouragement, highly supportive recommendation, and thought-provoking suggestion. We also appreciate the kind recognition of our in-house-developed cascade reaction. Below, I will respond to the referee's suggestions one by one.

1. It would be nice if the authors could comment, if they have tried to perform the pivotal coupling intramolecularly.

Response: Thank you for your suggestion. Yes, we have indeed performed numerous intramolecular Giese-type coupling reactions to construct the quaternary carbon at the C10 position. Previously, we included this work primarily in the Supplementary Information. Based on the suggestions from Referees #2 and #3 (the 3rd point), we have now incorporated this section into Figure 4e (as shown below).

Moreover, we also made the following comment in the main text: “We also investigated the intramolecular Giese reaction using acids **34–36** (Figure 4e, see Supplementary Fig. 27–29 for their preparations) and the RAEs thereof **37–39** (see Supplementary Fig. 31 for their preparations); unfortunately, we could not detect the desired coupling products. We speculated that the failures were due to: i) the C1-ester groups not favoring the *cis* configuration needed in the reaction transition state,

and ii) the relatively high barrier of rotation around the ester C–O bond to the desired reaction transition state⁵⁵. Furthermore, attempts to use the C10-acyl telluride as the tertiary radical precursor⁵⁶ also proved unsuccessful in both intermolecular and intramolecular Giese reactions (see Supplementary Fig. 26 and 33 for details).”

Reaction transition states: sterically bulky *cis*-esters are unfavorable

2. Also, I would re-draw the Figure 1 as the successful synthesis is different from what was planned initially. The paper should not reflect chronological how the research was preformed but how the final synthesis looks like. In Figure 1 the reader is put on a track which was not successful.

Response: Thank you for your thoughtful suggestion regarding Figure 1. I truly appreciate your perspective. While I acknowledge that traditionally, the retrosynthetic analysis might appear to represent the final synthesis, especially in a concise Communication-type article, I believe that in our full Article, presenting both the initial plan and the successful synthesis offers valuable context and insight into our research process. This approach allows readers to understand the challenges we faced and the adjustments we made along the way. It helps to illustrate the logical progression, problem-solving aspects, and the evolutionary nature of our studies, which we consider important for a comprehensive understanding of our work. Additionally, considering the positive feedback from Referee #3, who stated that our manuscript "is well written," we have decided to retain the current approach. We hope you understand our rationale for this decision and we sincerely appreciate your decision that this choice “should not deter the paper from publication”.

3. In the experimental, the alpha-D-value should also be provided for the intermediates.

Response: We appreciate the Referee for this kind reminder. The alpha-D-value have now been provided for every chiral intermediate in the Supplementary Information.

Finally, we again thank the Referee for the carefully considered and extremely helpful comments and guidance about improving our study. We are sincerely grateful for your thorough review and constructive feedback, which have undoubtedly strengthened our manuscript.

Reviewer #3 (Remarks to the Author):

The author reported the first and enantioselective total syntheses of C-19 oxygenated enmein-type *ent*-kauranoids, including (–)- macrocalyxofornins A and B, and (–)-ludongnin C. Key feature of the synthesis is a newly devised Ni-catalyzed reductive decarboxylative cyclization/radical-polar crossover/*C*-acylation cascade that forges a THF ring concomitantly with the β -keto ester group. A notable aspect of this synthesis is the innovative Ni-catalyzed reductive decarboxylative cyclization/radical-polar crossover/*C*-acylation cascade, which enables the formation of a THF ring concurrently with the β -keto ester group. Detailed investigation into the reaction mechanism of this cascade reveals that the *C*-acylation process involves carboxylation followed by in situ esterification. The manuscript is well written and the literature references are properly cited, and this reviewer supports the acceptance of this manuscript after minor revisions.

Response: First, we would like to extend our heartfelt gratitude to Referee #3 for their thorough review and encouraging feedback.

A few suggestions are listed as follows:

1. The isolated yield of the decarboxylative cyclization/RRPCO/*C*-acylation cascade of **11d** is moderate. It would be helpful to know if any byproducts were identified and characterized.

Response: The isolated yield of **11d** does indeed fall within the moderate range. Upon thorough examination, we found that despite the overall yield being 49%, the reaction exhibited cleanliness as evidenced by TLC analysis. Our attempts to detect potential impurities, such as compound **18**, only yielded minimal amounts. Furthermore, considering the complexity of the cascade reaction involving at least 6 steps, each achieving an average yield of approximately 90%, it suggests a commendable level of efficiency. While it's plausible that some impurities may arise in each step, their quantities are likely minimal, posing challenges in their detection during the final analysis. We think this sheds light on the matter.

2. Regarding the proposed mechanism for the decarboxylative cyclization/RRPCO/*C*-acylation cascade (Supplementary Fig.14), the authors should comment on the possibility of a transesterification

reaction in addition to the decarboxylation process involving the carbanion **20** reacting with Boc₂O to form intermediate III.

Response: We thank the referee for the thorough review. Indeed, this step involves a transesterification reaction. We mistakenly labeled the color of released CO₂ in this step, which led to the referee's misunderstanding that compound **20** undergoes decarboxylation before reacting with Boc₂O to form intermediate III. In fact, we have demonstrated that the C10 carbanion adjacent to the carbonyl group cannot directly react with Boc₂O (see Supplementary Fig. 5). Therefore, this reaction is a transesterification between compound **20** and Boc₂O. We have now corrected the color labeling in the revised Supplementary Information.

3. In the supporting information, the authors attempted to build up a molecular skeleton through an intramolecular free radical cyclization. Although the target product was not obtained, the authors also mentioned the intermolecular approach. It is suggested that the authors summarize and reflect these outcomes in the main text together to provide a complete story.

Response: Thank you for this suggestion! We have now summarized our attempts on intramolecular radical cyclization in Figure 4e and reflect these outcomes in the main text as below: “We also investigated the intramolecular Giese reaction using acids **34–36** (Figure 4e, see Supplementary Fig. 27–29 for their preparations) and the RAEs thereof **37–39** (see Supplementary Fig. 31 for their preparations); unfortunately, we could not detect the desired coupling products. We speculated that the failures were due to: i) the C1-ester groups not favoring the *cis* configuration needed in the reaction transition state, and ii) the relatively high barrier of rotation around the ester C–O bond to the desired reaction transition state⁵⁵. Furthermore, attempts to use the C10-acyl telluride as the tertiary radical precursor⁵⁶ also proved unsuccessful in both intermolecular and intramolecular Giese reactions (see Supplementary Fig. 26 and 33 for details).”

4. The authors mentioned a good yield in the free radical cyclization reaction of substrate **38**. It is curious whether oxygen and other initiators need to be added during the operation, as the reaction is carried out in an argon atmosphere according to the supporting information.

Response: We appreciate the insightful comment from the Referee. Although the reaction was conducted in an argon atmosphere with the use of an argon balloon, we acknowledge that the initiation

step indeed requires trace amounts of O₂ according to the mechanism. It is likely that the trace amount of O₂ present in the argon balloon initiated the reaction. To address this concern, we have added a note in the revised Supplementary Information to clarify this aspect of the reaction procedure.

5. Page S40, regarding preparation of substrate S33, does the additional iodine in the system require reductive quenching? Is there anything worth noting?

Response: We again thank the referee for the thorough review and suggestions. Indeed, we used saturated aqueous Na₂S₂O₃ to quench the additional iodine. We have now revised our work-up procedure in the supplementary information to include this detail.

7. Both the main text and supporting information require further editorial modifications.

The third paragraph, page 10: “an” should be deleted, and “lactones” should be changed into “lactone”.

The first paragraph, page 11: “giving raise to” should be changed into “giving rise to”, and “override” should be changed into “overrode”.

The third paragraph, page S48: “provide” should be changed into “was provided”.

Page S49: the mass spectrometry data of product **25** needs to be left aligned.

Response: We sincerely thank the referee for their careful review and detailed suggestions. We have thoroughly addressed these errors.

Finally, let us again take this final opportunity to sincerely thank the Referee for the encouragement and the truly helpful guidance about how to improve both the rigor and scientific impact of our study.

REVIEWERS' COMMENTS

Reviewer #1 (Remarks to the Author):

The authors have carefully revised the article. I suggest publishing this article directly in Nature Communications.

Reviewer #2 (Remarks to the Author):

I am satisfied with the changes the authors made to the manuscript. They kept the drawing I recommended to be changed. However, I can follow their arguments and are happy now to recommend publication as it is.

Reviewer #3 (Remarks to the Author):

Since the authors have adequately addressed the concerns raised by this reviewer, publication in Nature Communications is recommended without further revisions.